# Glaciological characteristics in the Dome Fuji region and new assessment for "Oldest Ice"

Nanna B. Karlsson[1,*], Tobias Binder[1], Graeme Eagles[1], Veit Helm[1], Frank Pattyn[2], Brice Van Liefferinge[2], and Olaf Eisen[1,3]

[1]Alfred-Wegener-Institut Helmholtz-Zentrum für Polar-und Meeresforschung, Bremerhaven, Germany
[2]Laboratoire de Glaciologie, Université Libre de Bruxelles, CP 160/03, Avenue F.D. Roosevelt 50, 1050 Brussels, Belgium
[3]Department of Geosciences, University of Bremen, Bremen, Germany
[*]now at: Geological Survey of Denmark and Greenland, Copenhagen, Denmark (nbk@geus.dk)

*Correspondence to:* Nanna B. Karlsson (nanna.karlsson@awi.de)

**Abstract.** A key objective in palaeo-climatology is the retrieval of a continuous Antarctic ice-core record dating back 1.5 Ma. The identification of a suitable Antarctic site requires sufficient knowledge of the subglacial landscape beneath the Antarctic Ice Sheet. Here, we present new ice thickness information from the Dome Fuji region, East Antarctica, based on airborne radar surveys conducted during the 2014/15 and 2016/17 southern summers. Compared to previous maps of the region, the new dataset shows a more complex landscape with networks of valleys and mountain plateaus. We use the new dataset as input in a thermokinematic model that incorporates uncertainties in geothermal heat flux values in order to improve the predictions of potential ice-core sites. Our results show that especially the region immediately south of Dome Fuji station persists as a good candidate site for obtaining an old ice core. An initial assessment of basal conditions revealed the existence of what appears to be subglacial lakes. Further radar data analysis shows overall high continuity of layer stratigraphy in the region. This indicates that extending the age–depth information from the Dome Fuji ice core to a new ice-core drill site is a viable option.

## 1 Introduction

To better constrain the response of the Earth's climate system to continuing emissions, a better understanding of past climate change is essential. A key advance would be to understand the transition in the climate response to changes in orbital forcing during the 'mid-Pleistocene transition' (900 ka to 1200 ka ago). Marine records indicate that during this time the periodicity of the glacial cycles changed from 40 ka to our current 100 ka (e.g. Lisiecki and Raymo, 2005). The driver of this change is not well understood and in particular the role of the atmospheric $CO_2$ and other greenhouse gases is of great interest. Therefore a key goal in the ice-core community is to retrieve a continuous climate record of this transition since only ice cores contain the unique quantitative information about past climate forcing and atmospheric responses (Wolff et al., 2005). An Antarctic ice core extending 1.5 Ma back in time (termed the "Oldest Ice" core) would provide not only a local Antarctic climate history but also global greenhouse gas concentrations (Fischer et al., 2013). This would be key to unravel the linkages between the carbon cycle, ice sheets, atmosphere and ocean behaviour. However, so far continuous ice-core records that may provide essential evidence about past mechanisms of climate change more than 1 Ma ago, have not been retrieved.

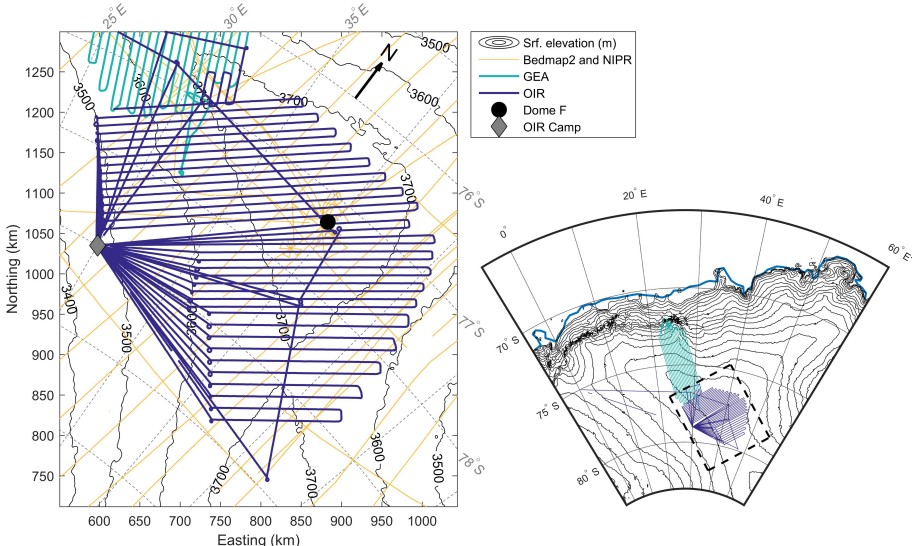

**Figure 1.** The Dome Fuji region in East Antarctica (the left-hand figure is a polar stereographic projection with standard parallel of -70°). The map shows the coverage of the new radar survey radarlines in 2014/15 (green) and 2016/17 (blue), and the yellow lines show the Bedmap2 data (Fretwell et al., 2013). Dome Fuji station and the OIR field camp are marked with a black dot and a grey diamond respectively. Surface contours from Bedmap2 are shown in black.

Under the umbrella of the International Partnerships in Ice Core Sciences (IPICS), the European "Beyond EPICA – Oldest Ice" (BE-OI) consortium and its international partners aim to retrieve an ice core up to 1.5 Ma old. As part of the pre-site survey plan, several extensive airborne operations have already been carried out. Campaigns have been conducted in the Dome C region, revealing several sites of interest and pinpointing areas for targeted exploration (Young et al., 2017), and extending
the known ages of the EPICA ice core to surrounding sites (Cavitte et al., 2016). Results from these surveys have provided additional constraints for modelling efforts to improve site predictions (Parrenin et al., 2017) or to estimate geothermal heat fluxes (Passalacqua et al., 2017). At Dome Fuji, two ice cores have been retrieved extending 330 ka and 720 ka back in time, respectively (Watanabe et al., 2003; Kawamura et al., 2017). The second deep ice coring project reached a depth of 3028 m where the ice is an estimated 720 ka and close to the melting point (Kawamura et al., 2017). However, in the wider region
around the ice core site (Fig. 1), the subglacial topography has so far been undersampled. The latest compilation of Antarctic ice thicknesses (Bedmap 2, Fretwell et al., 2013) relies mainly on Soviet airborne data from the 1970s with a large navigational uncertainty, and on ground-based Japanese surveys with a limited spatial coverage. Here, we present an updated ice thickness dataset for the Dome Fuji region including new data from two radar surveys carried out by the Alfred Wegener Institute, Helmholtz-Centre for Polar and Marine Research (AWI). In this region, there is a likely presence of old ice according to earlier
evaluations (Fischer et al., 2013; Van Liefferinge and Pattyn, 2013). We use the new dataset to update this prediction and issue recommendations for future field campaigns in the region.

## 2 Ice Thickness

### 2.1 Observations and Methods

The new topography relies primarily on datasets from two field campaigns conducted during the Antarctic seasons 2014/15 and 2016/17 (Fig. 1 green and blue lines, respectively). In both seasons the radar instrument was mounted on AWI's Basler BT-67 aircraft. The data from the 2014/15 season were collected as part of the GEA (Geodynamic Evolution of East Antarctica, see Ruppel et al. 2018) project, a collaboration between AWI and the Federal Institute for Gesosciences and Natural Resources, Germany. In that earlier survey, close to 40,000 km of radarlines were conducted but here we include only the 10 flights that directly intersect our area of interest corresponding to 12,000 km. The spacing between the radarlines is 10 km for the grid and the mean data-collection spacing along the flight direction is approximately 5 m although it varies with aircraft speed which is influenced by wind speed and direction. The 7-fold stacked data thereby get a spacing of approximately 35 m. The more recent survey is part of the Oldest Ice Reconnaissance (OIR) project, a contribution to BE-OI. During this field campaign, measurements were conducted from a temporary camp (located at 79°S, 30°E) 290 km from the Dome Fuji station. A total of 19,000 km of radar data were collected in 26 radarlines. The spacing between the OIR radarlines is 10 km with the exception of the radarlines southeast of Dome Fuji that were flown with a 15 km spacing, and the lines acquired while leaving or approaching the OIR camp where the distance between radarlines is much smaller. The mean data-collection spacing along the flight direction was also approximately 5 m leading to the same spacing on the 7-fold stacked data as the GEA data, i.e., 35 m, The radar data in both campaigns were collected using AWI's EMR (Electromagnetic Reflection) system (Nixdorf et al., 1999) with a constant distance to the ice surface. Radar waves were emitted with a centre frequency of 150 MHz and an amplitude of 1.6 kW as a 600 ns long pulse aiming to return a clear signal from the ice/bedrock interface as well as capturing information on the englacial properties of the ice. The system rectifies the returned energy and applies a logarithmic amplification.

To determine the ice–bed interface, only moderate processing was applied to the data, mainly 7-fold horizontal stacking and modest filtering to decrease noise. The bed returns were picked manually using the seismic software package ECHOS (2014/15 data) or semi-automatic detection routines developed in MATLAB (2016/17 data). The surface returns were automatically determined from the radar altimeter-reading simultaneously operated on the plane, filtered for outliers and smoothed. Subglacial lakes and locations of basal melt were identified based on a manual assessment of the reflection strength of the basal signal. For the calculation of the internal layer continuity index (cf. Karlsson et al., 2012) the logarithmic value of the stacked data was used. The top and bottom 20% of the ice column was discarded in the calculation to avoid surface noise and the reduced signal from the deepest part of the ice.

In addition to the two datasets described above, the new ice thickness dataset also includes the data from the Japanese and German surveys that are part of the Bedmap2 compilation. We have not included the Soviet data from the Bedmap2 data in our compilation because the data were collected without Global Positioning System information and thus have a high associated uncertainty in location (pers. comm., Fujita 2017, see also Lythe et al. 2001). The ice thickness was constructed in the following way: The difference between the surface and the bed signal was converted from two-way travel time to distance assuming a signal velocity in ice of $1.67 \cdot 10^8$ m/s with a firn correction of +10 m following Fretwell et al. (2013). We assume that the

thickness has not changed between time of data acquisition – a reasonable assumption given that elevation changes at Dome Fuji are less than 0.25 m a$^{-1}$ (e.g. Helm et al., 2014), thus the observed elevation changes are below the known accuracy of the radio-wave propagation speed. The data were subsequently regridded to a 500 m and a 1 km resolution grid using a kriging interpolation scheme: Based on the experimental variogram, the lag is set to 80 km. The experimental variogram is then fitted to an exponential model whose parameters are found by minimising the mean squares difference between the observational variogram and the model. The result of the kriging – the gridded ice thickness – is then merged with the Bedmap2 topography. A weighed mask was constructed wherein grid points more than 50 km from our survey points were assigned the Bedmap2 value, and grid points less than 20 km from our survey were assigned values from our newly interpolated data. Finally, the grid points in-between were assigned a linear combination of the two datasets with decreasing weight on OIR data with increasing distance from the OIR data points. For the 500 m resolution grid, that Bedmap2 dataset was first interpolated to a 500 m grid using nearest neighbour interpolation and then merged with the new data.

## 2.2 Results and Uncertainties

The resulting ice thickness is displayed in Fig. 2 and in the following we refer to this dataset as the OIR (ice thickness) data although it is based on data from several surveys (cf. the radarlines shown in Fig. 2A). The denser data coverage from the GEA and OIR surveys reveals a landscape with ice thicknesses varying between 2000 m and 4000 m with an average ice thickness of 3021 m (Fig. 2B). Immediately south of Dome Fuji is an area with shallow ice thicknesses while the ice further to the south and to the southeast and is significantly thicker. A complex terrain with a patchwork of thick and thin ice has now become visible north and west of Dome Fuji. Since the surface topography in this part of Antarctica is relatively flat, ice thicknesses are good indicators of bed topography, thus thick ice indicates valleys while shallower ice indicates mountains or highlands. The OIR dataset shows a system of valleys surrounded by high plateaus.

The difference between the OIR and the Bedmap2 datasets is shown in Fig. 2C. The largest differences appear to the west of Dome Fuji between 30°W and 35°W, and 77°S. In the immediate vicinity of the station, differences are smaller due to the high-resolution data from Japanese ground-based surveys that were included in the Bedmap2 and the OIR compilations. The difference between the new ice thickness map and Bedmap2 is upwards of 800 m in some areas, with a mean difference of 20 m, a mean absolute difference of 136 m, and a standard deviation of 177 m. This difference is undoubtedly due to the geolocation uncertainty of the Soviet data included in Bedmap2.

We examine the uncertainties of our result by comparing the gridded ice thickness map with point measurements from the individual radarlines, and through crossover analysis of the radarlines. In the latter analysis, we consider all points within 50 m of each other to be crossover points following Fretwell et al. (2013). For the GEA-OIR radarlines, 2453 points were found to be crossover points. The mean difference in ice thickness between the radarlines is -13 m with a standard deviation of 79 m. For 18% of the crossover points the difference exceeded 100 m. We ascribe this difference to geometrical effects of flightline orientation since we observe that radarlines intersecting each other at oblique angles have a larger thickness difference than those that are almost parallel. Of the crossover points where the difference exceeds 100 m, one third is situated along the radar line that runs from Dome Fuji to the west, intersecting the other radar lines.

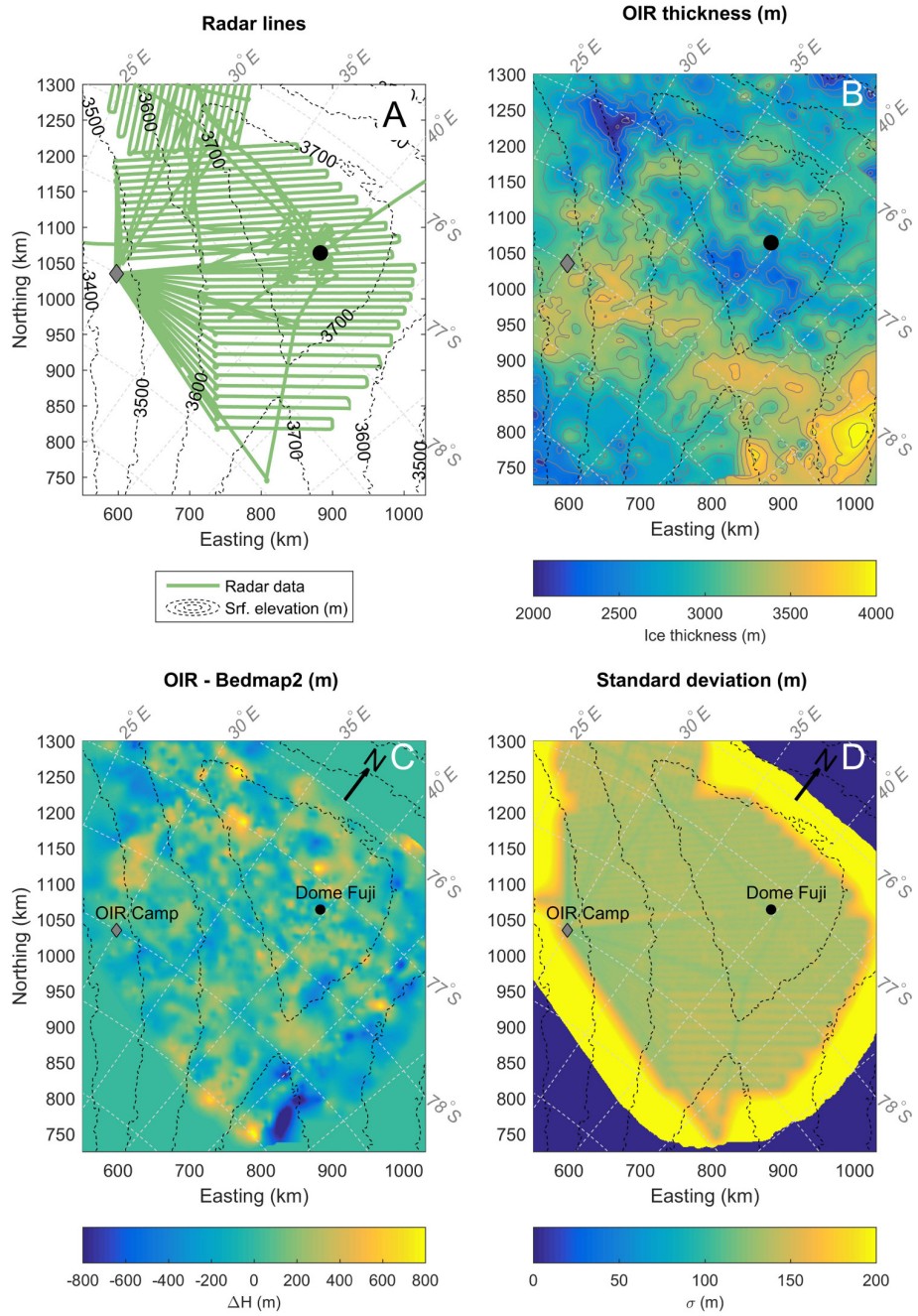

**Figure 2.** (A) The location of the radarlines from which the OIR dataset is constructed. (B) OIR ice thickness on grid (1 km horizontal resolution). (C) The difference between the OIR ice thickness grid and the Bedmap2 ice thickness. Positive values indicate that the OIR data show larger ice thicknesses in an area. (D) The standard deviation of the kriged OIR ice thickness. In all maps Dome Fuji is indicated with a black dot and OIR camp with a grey diamond.

Turning to the gridded ice thickness map, the mean difference between the GEA-OIR point measurements along the radarlines and the gridded map and is 1.5 m (cf. Fig. 3), indicating that the gridded ice thickness is slightly underestimating the ice thickness on this spatial scale. The standard deviation is 117 m. The mean difference between the gridded ice thickness and the individual data points from the older German survey is -11 m and the standard deviation is 162 m. For the Japanese surveys the mean difference and standard deviation is -16 m and 187 m, respectively. The overall high value for the standard deviation is an inevitable result of the smoothing introduced by the kriging interpolation scheme. Indeed, the negative mean difference between the high-resolution ground-based survey data and the gridded map indicates that the gridded data tend to overestimate the ice thickness in some areas, most likely because subglacial valleys are made broader by the kriging. This is a direct result of the smoothing introduced by the kriging scheme. This is also evident from the standard deviation in the grid points calculated by the kriging scheme (Fig. 2D). The standard deviation lies typically between 110-120 m close to the data points and increases with distance. Towards the margin of data coverage area the values exceed 200 m. We note that 110–120 m is still smaller than the difference between the OIR data and Bedmap2.

The decision to exclude the Soviet data was partly based on results from a crossover analysis. Comparison between the Soviet radarlines and the OIR and GEA surveys shows that only slightly more than 100 points is within 50 m of each other. For these points, the mean difference in ice thickness between the points is -5 m with a standard deviation of 193 m. This larger standard deviation is likely due the poorly resolved bed rock and the large navigational uncertainty in the Soviet radarlines. Considerable differences were also found between Soviet data and ground-based JARE surveys (pers. comm., Fujita 2017).

We performed a similar crossover analysis of the surface elevation measured in the GEA-OIR surveys. Here, we find a mean difference of less than 1 m with a standard deviation of 2.5 m. Thus, the uncertainties in the surface reflection measurements are negligible compared to uncertainties in the bed picking. Based on this we assign an average uncertainty of 150 m to the OIR ice thickness although we note that uncertainties increase with distance from data point are likely larger along the edge of the survey grid.

## 3    New Prediction of Oldest Ice Locations: Method and results

We apply the one-dimensional thermokinematic model described in Van Liefferinge and Pattyn (2013) to the OIR data. The model calculates the minimum required geothermal heat flux that will cause the bed to reach the pressure melting point. It is based on the simplified model of Hindmarsh (1999), wherein the one-dimensional temperature equation is solved while neglecting horizontal advection caused by ice-flow and further assuming steady-state conditions.

$$\kappa \frac{\partial^2 T}{\partial z^2} - w \frac{\partial T}{\partial z} = 0 \quad , \tag{1}$$

where $w$ is the vertical velocity, and $\kappa = K/(\rho c)$, where $K$ is the thermal conductivity, $\rho$ is the density and $c$ the heat capacity of ice. This approach neglects several processes that may influence the resulting geothermal heat flux, including changes over time in ice thickness or flow regime and horizontal advection. This is in particular problematic for areas that have experienced higher velocities than what is currently observed. For our study domain, ice thicknesses are thought to have varied between

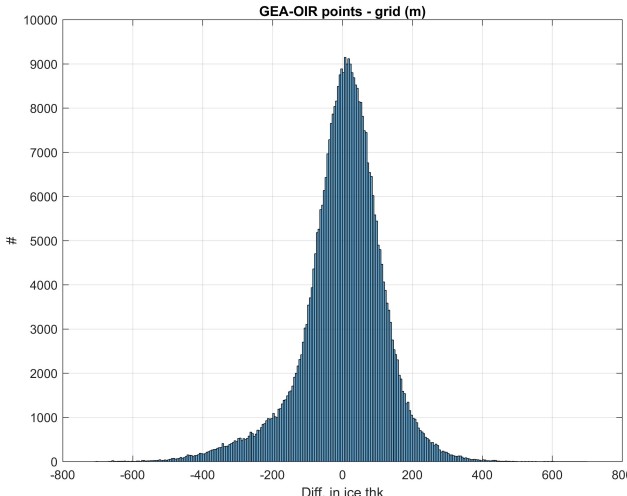

**Figure 3.** Histogram showing differences in ice thickness between the point measurements from the GEA-OIR surveys and the final gridded 1 km ice thickness data.

50 m and 250 m (Pollard and DeConto, 2009) and are thus unlikely to have experienced substantial changes in flow regime. In the original study by Van Liefferinge and Pattyn (2013), the model was applied to the entire Antarctic Ice Sheet on a 5 km resolution grid. The input parameters were horizontal ice velocities (assumed to be equal to balance velocities in our region of interest, see suppl. Fig. 7), surface mass balance (van de Berg et al., 2006; van den Broeke, 2008) and surface temperature (van den Broeke et al., 2006). The balance velocities were obtained from the balance fluxes, assuming that the mass of ice flowing out of any area exactly equals the sum of the inflow and the mass accumulated over the area (e.g., Budd and Warner, 1996). The vertically-averaged horizontal velocities thus depend on ice thickness and the prescribed mass balance. The ice is assumed to be internally deforming according to Glen's flow law with exponent n = 3 (cf. Pattyn, 2010). Note that the balance velocities are based on the old ice thickness data. The geothermal heat fluxes are from three different studies (Shapiro and Ritzwoller 2004; Fox Maule et al. 2005 and Purucker 2013). In our study, all above-mentioned parameters with the exception of ice thicknesses are identical to the fields used in Van Liefferinge and Pattyn (2013) but regridded to 1 km and 500 m resolutions using nearest neighbour interpolation. Thus, the difference in ice thickness between Bedmap2 and the OIR data is the main cause for the difference in results. As all the input parameters are smoothly varying on these spatial scales, we also expect a smooth minimum geothermal heat flux map especially since the geothermal heat flux datasets have a sparse special resolution of around 100 km. This implies that while the large-scale pattern is robust, detailed interpretation of features that are on a scale smaller than a few kilometres is not realistic.

The result from the calculation of the minimum geothermal heat flux is denominated $G_{min}^{pred}$. It is compared to the three different geothermal heat flux datasets, as well as $G_{mean}^{data}$ the average of the three datasets and $\sigma_G$ their standard deviation. The difference $\Delta G$ is then defined as

$$\Delta G = G_{min}^{pred} - G_{mean}^{data} \quad .$$

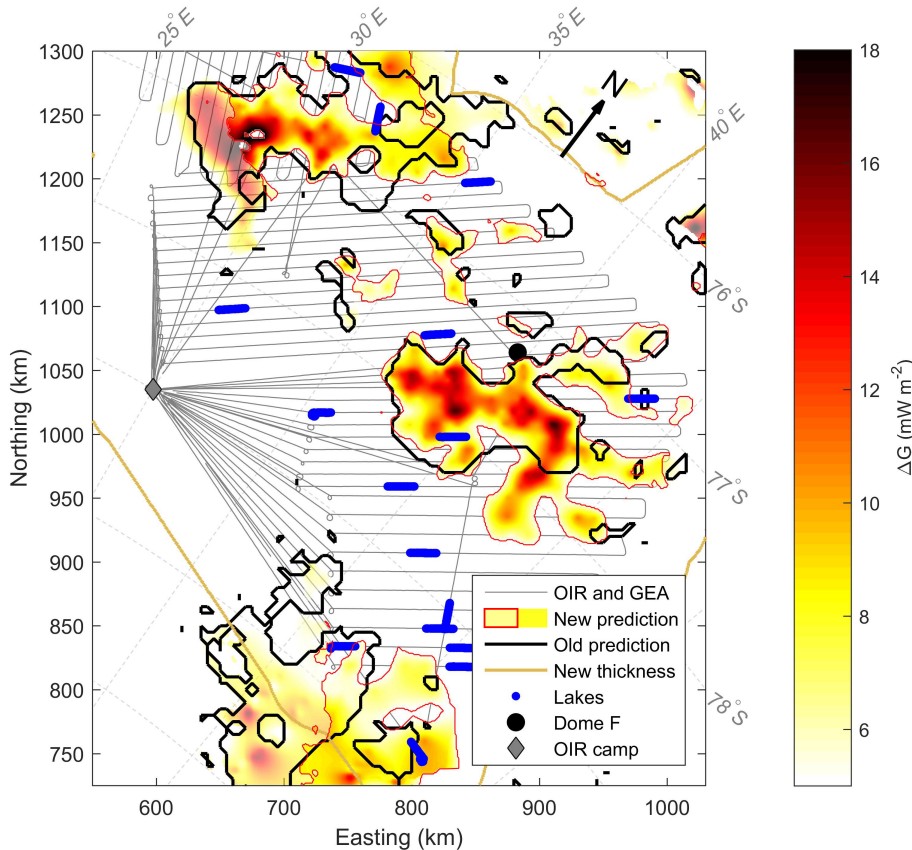

**Figure 4.** Updated predictions of possible Oldest Ice locations in colours compared to the prediction of Van Liefferinge and Pattyn (2013) outlined in black. The colour scale show the values of $\Delta G$, semi-transparent colours show areas with a threshold horizontal ice-flow velocity of <2 m/a, fully saturated colours show areas where the threshold is <1 m/a. Blue dots show lake locations identified from the radar data, the extent of the gridded OIR thickness map is outlined with a yellow line, i.e., predictions outside the line are based on Bedmap2 values (cf. Fig. 2B).

Potential candidate sites for Oldest Ice are assumed to be areas where $\Delta G$ is larger than 5 mW/m$^2$ and the standard deviation between the different geothermal heat flux datasets is low ($\sigma_G < 25$ mW/m$^2$). These two threshold values are chosen according to mean geothermal heat flux values for the whole interior Antarctic Ice Sheet. The values restrict the dispersion of geothermal heat flux values between data sets and reduce the likelihood of melting. The number of sites is further constrained by only

5  considering areas where the ice thickness is above 2000 m and the horizontal ice-flow velocities are <2 m/a. We refer readers to Van Liefferinge and Pattyn (2013) for an in-depth discussion about the choice of these parameter values.

The result of the model applied to the OIR data is shown in Fig. 4 in colours with black lines outlining the prediction of Van Liefferinge and Pattyn (2013) using Bedmap2. The colours of the figure show the value of $\Delta G$. Larger values of $\Delta G$ indicate that based on current observations it is likely that the geothermal heat flux is so low that the temperature of the bed is below

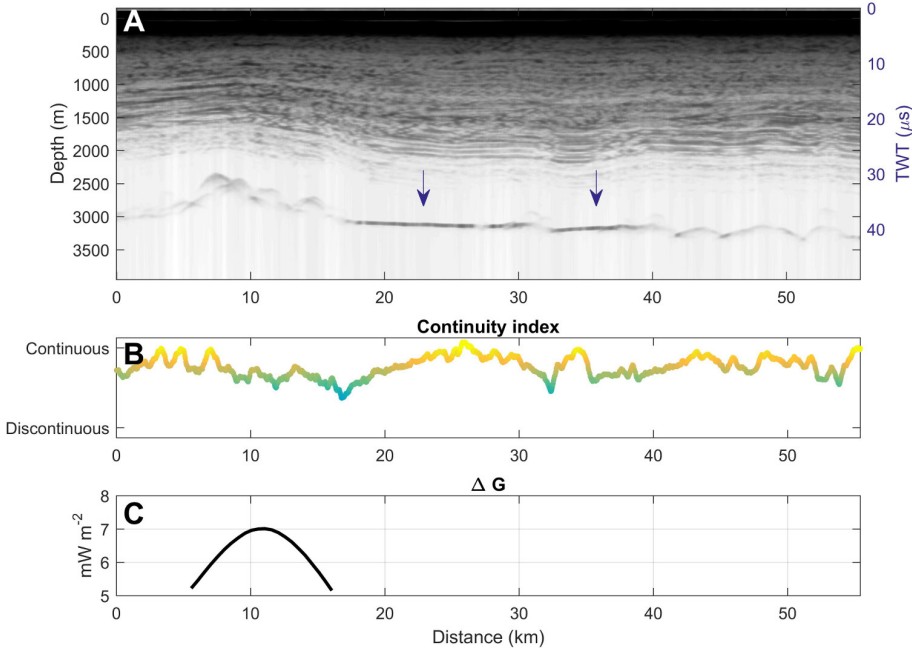

**Figure 5.** (A) Radargram (Z-scope) with two clear lakes indicated with blue arrows. The location of the radargram is indicated on Fig. 5 (B) Continuity index along the radargram. (C) Calculated $\Delta G$ assuming a maximum allowed horizontal velocity of $<2$ m/a.

the pressure melting point. Note that the old prediction is based on both the simple model presented above and results from an ensemble of runs with a more advanced three-dimensional ice-flow model. The new prediction roughly outlines the one based on Bedmap2. The areas of high likelihood of Oldest Ice fall into the same three areas: one approximately 250 km to the west of Dome Fuji, one in a large sector immediately south of Dome Fuji and one approximately 200 km further to the south. Notably,

the area to the west has decreased in size and areas that were considered likely to contain old ice are now no longer probable candidates for the chosen conditions. The two other areas have increased slightly in size. However, repeating the ensemble runs with the OIR dataset might modify this prediction but that is beyond the scope of this study.

Given that the thermokinematic model only depends on local parameters, the difference between our results and those presented in (Van Liefferinge and Pattyn, 2013) reflects the updated ice thickness information since all other parameters are identical.

Thus, our results are sensitive the uncertainty in ice thickness.

## 4    Basal conditions and internal layering

In order to make the best informed decision on the optimum drill site for retrieving Oldest Ice, we now return to the radar data for additional information. Based on a manual assessment of basal signal reflection strength and specularity of the signal (cf., Siegert et al., 2005), we have identified several areas that exhibit signs of the presence of subglacial water (blue dots, Fig. 4).

In the following we refer to these areas as "lakes" although in our preliminary analysis it is not possible to distinguish between

a lake or, for example, water-filled sediments. An example of a radargram with two clear lakes is shown in Fig. 5 along with the corresponding layer continuity (cf. Fig 6B) and the value of $\Delta G$ from the model (cf. Fig. 4). Four additional radargrams showing examples of lakes and the layer continuity can be seen in the supplementary material. The figures also give insight into how the model resolves small-scale features. In Fig. S4, two small lakes separated by a topographic high are visible in the radargram. In the kriged ice thickness data, this topographic high is smoothed out and to the model the lakes therefore appear as one. Previous studies have also identified lakes in the region (Siegert et al., 2005; Fujita et al., 2012) but our analysis adds several new locations (Fig. 6A). A few of the lakes from the above-mentioned study do not appear in our dataset. This might be due to the uncertain geolocation from the old radar data or to the angle of interception of the new radarlines. The identification of these lakes is somewhat subjective and probably underestimates the amount of liquid water present at the bed. Assuming that the subglacial water follows the steepest gradient in the hydropotential (Shreve, 1972), we estimate the potential drainage routes of the water. Fig. 6A shows the hydropotential (coloured contours), subglacial lakes (black dots) and their drainage routes (blue lines). The updated Oldest Ice prediction is outlined in black. Although several lakes are in the vicinity of sites that may contain Oldest Ice, the water is draining away from these regions, indicating that subglacial water is not easily introduced to the sites.

One key priority for drilling for an Oldest Ice core is not just the existence of old ice but also a sequentially stacked and undisturbed ice column. The internal stratigraphy of the ice, as imaged by the radar data, may provide a valuable constraint. The radar data are therefore analysed for layer continuity using the method of Karlsson et al. (2012). This automatic method gives an indication of the continuity of the layers, i.e., how coherent and easily traceable they might be. The analysis was conducted only on the OIR data collected during the 2016/17 field season since the method relies on consistency in the radar system settings and processing chain in order to be comparable between radarlines. Fig. 6B shows the continuity index smoothed by a moving window of 100 horizontal samples ($\sim$3.5 km). Evidently, the layer stratigraphy southeast of Dome Fuji is markedly more continuous than layers north and west of the station. Indeed, the candidate sites for the Oldest Ice do not have lower layer continuity compared to some of the regions outside of the candidate sites. However, it should be emphasised that this does not preclude tracing of the layer. The continuity index merely indicates that there are potentially fewer well behaved layers, or that in order to resolve the layers a survey would be needed with higher vertical or horizontal resolution on the order of metres and kilometres, respectively, for example, by ground-based radar.

## 5 Implication for ground-based Oldest Ice surveys

Based on the OIR ice thickness dataset, we identify two regions with the best potential for containing Oldest Ice. Our criteria include modelled $\Delta G$ values, ice-flow velocities, proximity to the Dome Fuji ice-core and local ice thickness. For the latter criterium, we make a qualitative assessment essentially assuming that shallower ice thicknesses result in lower time-resolution but we make no quantitative analysis here. The most promising region is also the most easily accessible region from the existing Dome Fuji station: the region immediately south and southeast of Dome Fuji (black box, Fig 6B). For this region, we favour two areas outlined with white/black rectangles in Fig. 7, that each contain two local maxima where $\Delta G$ is elevated. Although

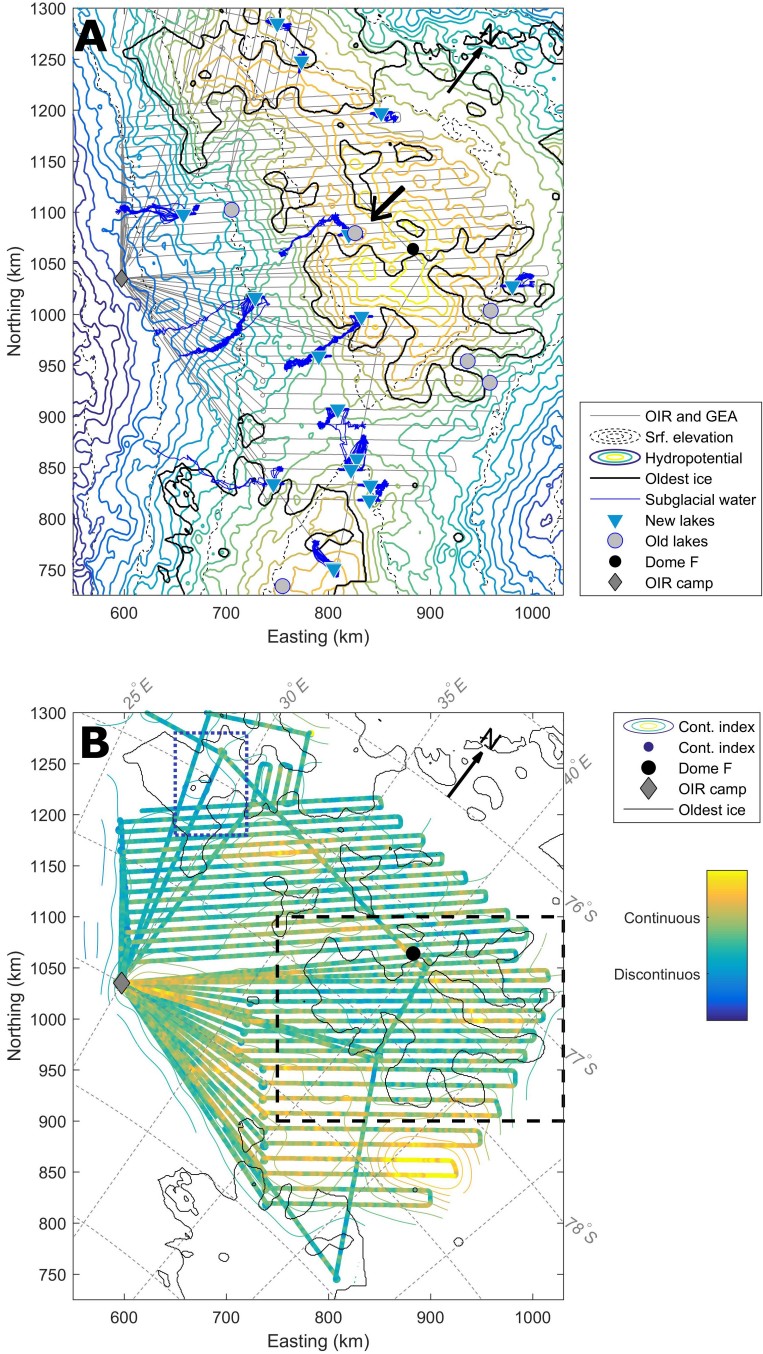

**Figure 6.** (A) The hydropotential (coloured contours) with the new Oldest Ice prediction outlined in black. The identified lakes are shown with triangles and the drainage routes of the water with blue lines. Grey circles indicate lakes idenftied in earlier studies (Siegert et al., 2005; Fujita et al., 2012). The black arrow indicates the location of Fig. 5 (B) Continuity index of the internal layers in the OIR radar data. The index has been smoothed with a horizontal window of 100 samples (∼3.5 km). Here, yellow colours indicate highest layer continuity. The dashed box shows the location of Fig. 7

the entire region has a value of $\Delta G > 5$ mW m$^{-2}$, we pinpoint the two areas in particular for two reasons: 1) The value of $\Delta G$ approaches and exceeds 12 mW m$^{-2}$, thus the uncertainty in geothermal heat flux is less important for the robustness of the prediction. 2) The bed topography appears to consist of high plateaus rather than rugged mountain terrain. The second reason is important for layer continuity and to avoid disturbances to the lowermost part of the ice, that would inevitably impact the stratigraphy. The proximity to the ice core site also implies that extending the age–depth information from the ice core to a new drill site would be relatively straightforward. Especially the area "II" where the distance is small and the bed topography relatively smooth is a promising site.

The second region with a potential for Oldest Ice is the site west of Dome Fuji (blue dotted box, Fig. 6). In the western part of this area, values of $\Delta G$ are also high, although at a first glance two properties make it less favourable: Firstly, the horizontal surface velocities approach and exceed 2 m a$^{-1}$ in this area, increasing the travel distance of the particles and thus making interpretation of an ice core more complicated; secondly, the ice is relatively thin in the region (typically less than 2.5 km) which may prove problematic for obtaining an adequate age resolution in an ice core. Furthermore, the distance (several hundreds of kilometres) to Dome Fuji makes it challenging to connect the internal layers to the ice-core chronology. Even so, layer tracing has been achieved across equal or longer distances in this part of Antarctica (e.g., Fujita et al., 2011; Steinhage et al., 2013), and a targeted campaign with radar systems optimised for layer clarity could address this issue.

To verify that the two regions could indeed provide a suitable drilling target for IPICS Oldest Ice objectives, we recommend first further investigations in these two regions. High-resolution radar measurements, ideally from the ground, are needed to identify the layer integrity especially in the basal region, i.e. the lowermost 20% of the ice. Temperature measurements in boreholes in the upper ~600 m of the ice column need to be merged with measurements of the vertical velocity by phase-sensitive radar systems (e.g., Nicholls et al., 2015). Using those datasets with ice-flow modeling and the age–depth distribution extrapolated from the Dome Fuji ice core would provide better estimates of the age near the bed as well as the respective annual layer thickness, which constrains the applicability of currently available ice core analytics (Fischer et al., 2013). This would provide further constraints to localize areas suitable for rapid access drilling (e.g., Schwander et al., 2014) to be deployed in a second step, which would enable a preliminary analysis of climate proxies and thereby constrain the age of the ice at the sample site by comparing directly with marine climate records.

## 6 Conclusions

A new ice thickness dataset for the Dome Fuji region has been constructed from airborne radar data with the aim to improve predictions of sites that may contain ice that is older than 1.5 Ma. The new data resolve the topography in substantially higher detail than previously published data, revealing a landscape of valleys and highlands. A manual assessment of the data also identified several areas exhibiting signs of the presence of subglacial lakes, and our analysis indicates that any subglacial water drains away from potential Oldest Ice sites.

We use the new data to force a thermokinematic model and update the prediction for candidate sites. Based on the model results, we presented a new assessment of areas in the Dome Fuji region where the presence of ice older than 1.5 mio. year is

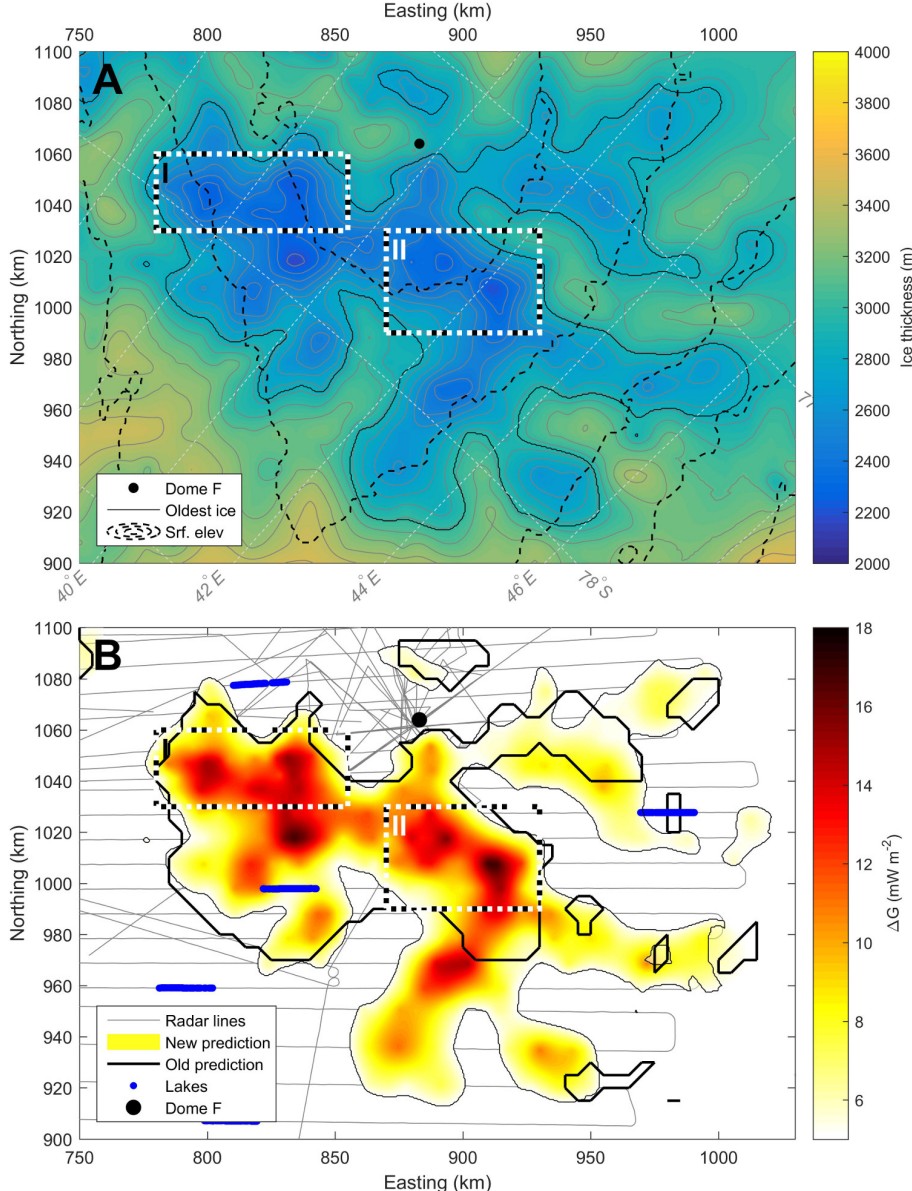

**Figure 7.** (A) Ice thickness in the area close to Dome Fuji station in 500 m resolution and (B) Updated predictions of possible Oldest Ice locations (colours) on a 500 m resolution grid compared to the prediction of Van Liefferinge and Pattyn (2013) outlined in black. The colour scale show the values of $\Delta G$, semi-transparent colours show areas with a threshold horizontal ice-flow velocity of $<2$ m a$^{-1}$, fully saturated colours show areas where the threshold is $<1$ m a$^{-1}$. Blue dots show lake locations identified from the radar data. The white/black boxes indicate the two most favourable Oldest Ice spots according to our analysis.

likely. We identified two regions where the available margin for geothermal heat flux uncertainties is large enough to sustain old ice over several glacial-interglacial cycles. One such region is south of the Dome Fuji station, and within this region especially two areas are of substantial interest. We recommend further targeted investigations to these areas to ascertain layer continuity and to establish approximate age–depth information.

The gridded thickness data and the thickness picked along the radarlines will be made available through the pangaea.de website or upon request from the main author.

*Acknowledgements.* This publication was generated in the frame of Beyond EPICA-Oldest Ice (BE-OI). The project has received funding from the European Union's Horizon 2020 research and innovation programme under grant agreement No. 730258 (BE-OI CSA). It has received funding from the Swiss State Secretariate for Education, Research and Innovation (SERI) under contract number 16.0144. It is fur-

10 thermore supported by national partners and funding agencies in Belgium, Denmark, France, Germany, Italy, Norway, Sweden, Switzerland, The Netherlands and the United Kingdom. Logistic support is mainly provided by AWI, BAS, ENEA and IPEV. The opinions expressed and arguments employed herein do not necessarily reflect the official views of the European Union funding agency, the Swiss Government or other national funding bodies.

We thank K. Grosfeld and J. Sutter for insightful discussions. The comments and suggestions from K. Matsuoka and two anonymous re-

15 viewers improved the quality and readability of this manuscript and we are grateful for their insights. This publication also benefitted from discussions with colleagues at NIPR, Tokyo, Japan, in particular with S. Fujita, who also provided JARE data for comparison. We thank P. Fretwell, who provided information on the datasets that are included in Bedmap2, and S. Popov who shared the data from the Soviet Antarctic Expedition with us. We thank AWI's logistics field team, the CoFi team and the flight crew for support during the expedition. This is BE-OI publication number xxx.

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
