# Peer review of "Glaciological characteristics in the Dome Fuji region and new assessment for "Oldest Ice""

_The Cryosphere, 2017_

## Referee Comment (RC1) · Anonymous Referee #1 · 9 Jan 2018

This paper present a new map of the bedrock topography in the Dome Fuji region, East Antarctica, using recent airborne radar surveys. The new map is used (i) to update the study of Van Liefferinge and Pattyn (2013) and infer areas that are probably not currently at the pressure melting points; i.e. potential areas to find ice million year old ice; and (ii) estimate the potential drainage routes of identified wet areas. A continuity index of the internal layers seen in the latest radar data is shown to estimate the potential of tracing layers from the Dome Fuji ice core location to identified Oldest Ice location; However this part is not very conclusive.

This new data set will be useful to drive further geophysical investigations in the area

to refine potential locations for Oldest Ice.

However, I think few points require more consideration from the authors:

- The title and conclusion claim that this study is a new assessment for 1.5 Ma old ice. The 1.5 Ma is a target of the Oldest-Ice project, but the study does not demonstrate that ice as old as 1.5 Ma could be retrieved in the area. The fact that the basal ice may not be at the pressure melting point for the present day conditions is not a sufficient condition to retrieve ice older that is 1.5Ma old; this could be affected by basal disturbances, past melting conditions etc... Van Liefferinge and Pattyn (2013) speak of "suitable sites for the preservation of million year old ice". At this stage it is too early to say that it could be up to 1.5 Ma. This will have to be refined by further studies aiming are providing potential ages for the basal ice. In this prospect it would be useful here to recall the age of the ice at the bottom of the Dome Fuji ice core.
- The new thickness map: Kriging naturally comes with an estimate of the uncertainty. This estimate is lost when merging the new map with bedmap2. It seems that this merge is needed because they exclude the Soviet data and thus have no data outside of the new thickness data area. It seems that there is a high spacial resolution of the new thickness data, so that providing the grid with the new data and the uncertainty estimate (eventually masked to give data only at grid points where the closest distance is less than 20km) might be more useful for further investigation than this merged map where the uncertainty information has been lost. It is not clear if the gridded data really include the old German and Japanese data, as the raw individual points are compared to the gridded product (page 4, lines 9-12)?
- New Prediction of Oldest Ice Locations: because they use a 1D thermokinematic model, all others parameters being identical to Van Liefferinge and Pattyn (2013), the solution depends only on the local ice thickness and not on the topography,
i.e. the thickness gradient. So for oldest ice locations, we should have differences between the old and the new studies only if areas are significantly thicker or shallower; which seems not to be the case here; there is only a better description of the topography and there is no large areas with significant differences with bedmap2. So it seems not surprising that there is a good agreement between the old and new map. Moreover, I am a bit concerned by the validity of the 1D assumption when the model is run with a horizontal spatial resolution (here 500m) much lower than the ice thickness. A more in-depth discussion on the limits of this approximation to study potential local variations is required and avoid the risk of over-interpreting the results.

Minor points:

In all Figures in stereo-polar projection have the longitude the labelled "W". It should be "E"? In Fig. 1: make the symbols bigger for Dome Fuji station and OIR camp. Fig.2 : It could be interesting to show some surface elevation contours. Fig.3 : The color scale shows the full range from 0; but in the figure, a threshold has been applied to show only values > 5 mW/m2? Page 9, line 9; "(white arrow, Fig. 5)". There is no white arrow in Fig. 5.

TCD

---

## Referee Comment (RC2) · K. Matsuoka (Referee) · 13 Feb 2018

This manuscript reports the new bed topography and associated predicted locations of the oldest ice in the Dome Fuji region. The data presented here have a high value; it is the first modern regional airborne surveys that are necessary to characterize this region. In this regard, this paper will be a main reference point for future work in this region and is worth to be published. However, in my opinion, this manuscript doesn't present full information package and individual conclusions are weakly supported by the analysis. I would like to see more comprehensive descriptions of the methods and data so that one can evaluate the work and utilize the results in an appropriate way. Therefore, I would recommend a major revision and I hope that my comments below are helpful.

1. The main shortcoming of this paper is incomplete description of the survey, data processing, and data properties. Main lacking information is:
    a. Survey: line spacing (mean, or range/max/min, it's hard to read from the figure), flight mode (constant elevation or constant clearance to the ice), nominal data-collection spacing along survey profiles.
    b. Data gridding: see #2 below.
    c. Lake detection: see #3 below.
    d. Continuity index: depth range analyzed for the index, see #5 below too.
2. The presented ice thickness map could include many artifacts due to unrealistically small grid size and smoothing to a large degree. Is the grid size of 0.5 km justifiable? Radar profiles are separated by > 5-10 km, and the authors admit that the large standard deviation (142 m) between gridded values and previous German/Japanese radar data is inevitable result of the smoothing (P4L11). I am afraid that this grid size gives a false impression that the bed topography is known better than we actually know. No details are given about smoothing or data interpolation. This is a key aspect of the final product presented here, however.
3. Very little information is given about lake identification. Nobody can reproduce or assess author's arguments in a meaningful way. Radargrams of at least several lakes should be presented. Do they look like definite lakes, or more like dim lakes? Are all known lakes along the new radar profiles found in this study?
4. Sections 2.2 and 2.3 should be better organized. Section 2.2 are written in a way to examine uncertainty of the new data but also to examine old data, which is confusing for me. Section 2.3 presents spatial patterns of the difference between OIR and BEDMAP2, which can be better presented if Sections 2.2 and 2.3 are more efficiently organized/integrated.
5. Continuity index presented in Fig. 5 should be shown only along the radar profiles and should not interpolate between profiles.
6. It is necessary to show radargrams that represent several typical features in this region (e.g., region with high continuity index, lake, transition from the cold base to thawed base) together with continuity index, lake appearance, Delta G, etc. The manuscript shows only limited aspects of the data properties and it is hard for readers to make a comprehensive assessment.
7. One of main conclusions is that the most promising sites for the oldest ice is in the box I and II. I agree with the authors about this conclusion in general but data should be better presented to fully support this argument. First, why do the authors recommend two separate regions Box I and Box II, instead of one large box that includes both I and II? Second, what defines the

boundaries of these boxes? These two are just examples showing the lack of full argument of the main conclusion.

8. Please enlarge figures, and consider grouping some figures into a single figure (panel a, b, c, ….). it's hard to compare several data properties together in the current form.
9. Mention data availability.

More detail comments:

Figure 1: include surface topography (with contours or background color) in both panels. I recommend enlarge both panels for clarity.

Figure 2: it's hard to distinguish fully saturated colors and half-transparent. Please modify the figure.

Figure 3: The candidate sites should have Delta G > 5 mW/m2. Is it better if the colorbar shows Delta G between 5 and 18 (instead of 0 and 18)?

Figure 4: Consider larger intervals of the contours. Many features are smaller than the radar-profile separations and thus could be artifacts. It's hard for me to see water paths.

P3L22: "Echo-free zone" is defined as the deep ice on which radar returned power decreases in a short vertical distance much more than expected with geometric spreading and englacial attenuation (Fujita et al., 1999). The echo-free zone is not identical with a zone that shows no return signals, which can be caused by limited radar performance.

P3L25: Is this true? I think that it is the case, but there is no evidence. Probably better to say something like "because the data were collected in late 1980s without modern GPS positioning system."

P3L29: Pulse widths determine a vertical resolution (capability to distinguish two objects separately) but irrelevant to the precision of the range measurement. The observed elevation change is about 0.01% of a typical ice thickness in the study area (2 km), which is below the known accuracy of the radio-wave propagation speed.

P4L5: show the number of cross-over points analyzed here.

P4L3: please mention clearly at the beginning of this paragraph "the uncertainties are analyzed by cross-over analysis of the new datasets and by the comparison of the gridded map product with previous data that were not used in the map product." Many uncertainty analyses are presented here and it's hard to see the overall analytical setup.

P4L8-9: consider showing a histogram of cross-over errors. Also, it makes more sense to show first and third quartile, instead of the standard deviation.

P4L10: unclear what negative number means here (which data were subtracted from which).

P4L11: flightlines? Confusing because JARE data are ground based.

P4L13-19: Comparison with the BEDMAP2 data shows large discrepancy, and the authors argue that it mainly caused by poor positioning of the earlier Soviet data (again: no evidence of the poor positioning). So, these statements are not for determining uncertainties of the newly collected data.

P4L24: characteristics of what?

P5L15: include the mean and range of geothermal flux in the study area taken from these datasets. I believe that the geothermal flux is a predicted field of this model (rather than an input field) and these geothermal flux datasets are used to derive delta G. So, I think that geothermal flux is mentioned at a wrong place.

P6 Equation: it's confusing to see $G\_min$ and $G\_mean$, as $G\_min$ is a model prediction and $G\_mean$ is the mean of three datasets. How about $G\_min(predicted)$ and $G\_mean(datasets)$ or such?

P6L2: Why is $Sigma\_G < 25$ mW/m2 considered as low? Geothernal flux in this region is roughly 50 mW/m2, so I understand that the standard deviation of 25 mW/m2 means that the geothermal flux has +/- 50% uncertainty. In other words, the margin of geothermal flux with which the bed is kept frozen is much less than the geothermal flux dataset uncertainty (i.e. Delta G of 5 mW/m2 is much smaller than the range of the dataset, 25 mW/m2). I understand that the full description of this modeling is given in Van Liefferinge and Pattyn (2013) but still key information should be given here.

P7L10: don't use several words in interchangeable ways; Lakes, wet areas, water-filled areas, ….

---

## Referee Comment (RC3) · Anonymous Referee #3 · 21 Feb 2018

This manuscript provides a new glaciological information (ice thickness, bedrock, topography, GHF, basal condition etc.) of the Dome Fuji area on the base on airborne radar surveys conducted during the 2014/15 and 2016/17 Antarctic season. An accurate geophysical survey is prerequisite for any paleoclimatic ice core site selection. The important effort to provide new geophysical survey in a remotest area of East Antarctica must be supported.

While the main result is of interest this manuscript suffers of some flaws: it is not clear the use of previous radar survey (Soviet, Japan and bedmap2 compilation) in the new map; the grid cell of 500 m is unsustainable and useless with a survey spacing

from 5 to 10 km and relative interpolation of tens km; the information about internal layering depth and age respect to Dome Fuij ice core are not reported anywhere; the description of subglacial lakes detection is very limited and the thickness of melting/no melting threshold is not reported, and it is a crucial point for GHF; as reported by Fisher et al., 2013 it is very important not only find undisturbed ice of 1.5 Myr but also that the ice between 1.2 and 1.5 myr must be enough thick to detect the 40 kyr cycle, a map of the deepest datable isochrones (from DF) is a crucial point to analyse and report. In detail:

Fig 1WGS84 is not a map projection, is a geodetic spatial reference system

All figure: add elevation contour line, change West with East, Are data plotted only OIR (fig 2 and 5) or GEA-OIR?

Page 3 Line 29-30 the elevation change is negligible information, remove.

Page 3 Line 29-33 If the Soviet data are not useful due to the position uncertainty why use bedmap2 based on this data? The new grid must be constructed using only the AWI and Japan data, and BEDMAP2 must be used outside the new survey area.

Paragraph 2.2 The paragraph must be clarify and analysed only the cross point of radar profile and not the gridded data.

Page 4 line 8 "parallel" or "perpendicular"?

Page 4 Line 16-19 what is the meaning of "points qualified"? The difference in ice thickness and standard deviation between Soviet and AWI does not appear so large respect to the comparison of GPS position survey of GEA-OIR and AWI-Japan. Please clarify the point and the uncertain in geographic position of Soviet profile.

Figure 2 Add the radar profile, why use only OIR profile, instead OIR e GEA? East instead of West

Figure 3 Elevation contour, yellow line New Thickness is very far from radar profile (25

km?), is it the gridded area?

Page 9 Line 2 Dome Fuji area is about 1000 km from the coast, and it is not "the most easily accessible region", but the closer to the Dome Fuji Station.

Fig 6 Add elevation contour line and radar line used, it is not clear (cf Fujita et al., 2012) explain.

---

## Author Response (AR1)

Note: Reviewer's comments are in italics. Changes to manuscript text are in blue.

Reply to reviewers:

**"Glaciological characteristics in the Dome Fuji region and new assessment for 1.5Ma old ice"**

**Reviewer #1**
**Main points**

**#1**

*The title and conclusion claim that this study is a new assessment for 1.5 Ma old ice. The 1.5 Ma is a target of the Oldest-Ice project, but the study does not demonstrate that ice as old as 1.5 Ma could be retrieved in the area. The fact that the basal ice may not be at the pressure melting point for the present day conditions is not a sufficient condition to retrieve ice older that is 1.5Ma old; this could be affected by basal disturbances, past melting conditions etc... Van Liefferinge and Pattyn (2013) speak of "suitable sites for the preservation of million year old ice". At this stage it is too early to say that it could be up to 1.5 Ma. This will have to be refined by further studies aiming are providing potential ages for the basal ice. In this prospect it would be useful here to recall the age of the ice at the bottom of the Dome Fuji ice core.*

Reply

That is a valid point. All mentions of 1.5Ma has been changed to "oldest ice".

The following lines have been added to include mention of the Dome Fuji ice core:

"Ice cores retrieved from Dome Fuji extend 330ka and 720 ka back in time (Watanabe et al., 2003, Kawamura et al., 2017). The second deep ice-coring project reached a depth of 3028 m where the ice is an estimated 720ka and close to melting point (Kawamura et al., 2017). However, in the region around the ice core site (Fig. 1), the subglacial topography has so far been undersampled."

**#2**

*The new thickness map: Kriging naturally comes with an estimate of the uncertainty. This estimate is lost when merging the new map with bedmap2. It seems that this merge is needed because they exclude the Soviet data and thus have no data outside of the new thickness data area. It seems that there is a high spatial resolution of the new thickness data, so that providing the grid with the new data and the uncertainty estimate (eventually masked to give data only at grid points where the closest distance is less than 20km) might be more useful for further investigation than this merged map where the uncertainty information has been lost. It is not clear if the gridded data really include the old German and Japanese data, as the raw individual points are compared to the gridded product (page 4, lines 9-12)?*

Reply

The new data are merged with bedmap2 only in areas where we have no new information. Thus, the uncertainty estimate from the kriging routine is still valid within the new thickness data area. We have

addressed this by including the calculated standard deviation from the kriging routine in Fig. 2. To clarify which lines are included in the gridded product, Fig. 2 now also includes a map of the radarlines. In other words, we have modified figure 2 so that it now includes A) a map of all the radarlines included in the final ice thickness product, B) the OIR ice thickness map, C) the difference between the OIR ice thickness and Bedmap2 and D) the standard deviation from the kriging routine.

We thank the reviewer for the suggestions on how to improve the readability and information in these sections. The section about results and uncertainties have been rewritten and now includes mention of standard deviation from the kriging and has a clearer outline of the uncertainty analysis. It now reads as follows:

"2.2 Results and Uncertainties

[revised manuscript text omitted]

**3**

*New Prediction of Oldest Ice Locations: because they use a 1D thermokinematic model, all others parameters being identical to Van Liefferinge and Pattyn (2013),the solution depends only on the local ice thickness and not on the topography, i.e. the thickness gradient. So for oldest ice locations, we should have differences between the old and the new studies only if areas are significantly thicker or shallower; which seems not to be the case here; there is only a better description of the topography and there is no large areas with significant differences with bedmap2. So it seems not surprising that there is a good agreement between the old and new map. Moreover, I am a bit concerned by the validity of the 1D assumption when the model is run with a horizontal spatial resolution (here 500m) much lower than the ice thickness. A more in-depth discussion on the limits of this approximation to study potential local variations is required and avoid the risk of over-interpreting the results.*

We agree that that the difference between the results from Van Liefferinge and Pattyn (2013) and the results presented here only reflect the new information about ice thickness. This is a result of the 1D model and its assumptions. We disagree that there are no areas that are significantly thicker or shallower. As presented in Fig. 3 we are able to exclude several areas as candidates for oldest ice sites. The area west of Dome Fuji (that has been considered to be an area of interest for some time) has been reduced in size, while the southernmost candidate site has changed position shifting almost 100km towards the east.

In response to the comments from other reviewers, we now present the model results on a 1km for our domain and on a 500m grid for the area around Dome F. We note that comparisons between results from 500m, 1km and 2km grids show small differences.

Additionally, we add the following to Section 3:

"[…] where $w$ is the vertical velocity, and $\kappa = K/(\rho c)$, where $K$ is the thermal conductivity, $\rho$ is the density and $c$ the heat capacity of ice. This approach neglects several processes that may influence the resulting geothermal heat flux, including changes over time in ice thickness or flow regime and horizontal advection.

This is in particular problematic for areas that have experienced higher velocities than what is currently observed. For our study domain, ice thicknesses are thought to have varied between 50m and 250m (Pollard and DeConto, 2009) and are thus unlikely to have experienced substantial changes in flow regime."

And

"In our study, all above-mentioned parameters with the exception of ice thicknesses are identical to the fields used in Van Liefferinge and Pattyn (2013) but regridded to 1km and 500m resolution using nearest neighbour interpolation. Thus, the difference in ice thickness between Bedmap2 and the OIR data is the main cause for the difference in results. As all the input parameters are smoothly varying on these spatial scales, we also expect a smooth minimum geothermal heat flux map especially as used geothermal heat flux datasets have a sparse special resolution around 100km. This implies that while the large-scale pattern is robust, detailed interpretation of features that are on a scale smaller than a few kilometres is not realistic."

**Minor points**

*In all Figures in stereo-polar projection have the longitude the labelled "W". It should be "E"?*

Yes, thanks for pointing that out!

*In Fig. 1: make the symbols bigger for Dome Fuji station and OIR camp.*

Symbols have been made bigger in Fig. 1 and 2.

*Fig.2 : It could be interesting to show some surface elevation contours.*

Surface elevation contours have been added to the new figures 1 and 2.

*Fig.3 : The color scale shows the full range from 0; but in the figure, a threshold has been applied to show only values > 5 mW/m2?*

The colour scale and its range have been changed for clarity.

*Page 9, line 9; "(white arrow, Fig. 5)". There is no white arrow in Fig. 5.*

Has been changed to a blue box.

New figure 2

[Figure]

New figure 3

[Figure]

Note: Reviewer's comments are in italics. Changes to manuscript text are in blue.

Reply to reviewers:

**"Glaciological characteristics in the Dome Fuji region and new assessment for 1.5Ma old ice"**

**Reviewer #2 (K. Matsuoka)**
**Main points**

**#1**

*The main shortcoming of this paper is incomplete description of the survey, data processing, and data properties. Main lacking information is:*

a. *Survey: line spacing (mean, or range/max/min, it's hard to read from the figure), flight mode (constant elevation or constant clearance to the ice), nominal data-collection spacing along survey profiles.*
b. *b. Data gridding: see #2 below.*
c. *c. Lake detection: see #3 below.*
d. *d. Continuity index: depth range analyzed for the index, see #5 below too.*

Response to 1a:

The following lines have been added in Section 2.1

"The spacing between the radarlines is 10 km for the grid and the mean data-collection spacing along the flight direction is approximately 5m although it varies with aircraft speed which is influenced by wind speed and direction. The 7-fold stacked thereby get a spacing of approximately 35m."

[…]

"The spacing between the OIR radarlines is 10 km with the exception of the radarlines southeast of Dome Fuji that were flown with a 15 km spacing, and the lines acquired while leaving or approaching the OIR camp where the distance between radarlines is much smaller. The mean data-collection spacing along the flight direction was also approximately 5m leading to the same spacing on the 7-fold stacked data as the GEA data, i.e., 35m. The radar data in both campaigns were collected using AWI's EMR (Electromagnetic Reflection) system (Nixdorf et al., 1999) with a constant distance to the ice surface."

**#2**

*The presented ice thickness map could include many artifacts due to unrealistically small grid size and smoothing to a large degree. Is the grid size of 0.5 km justifiable? Radar profiles are separated by > 5-10 km, and the authors admit that the large standard deviation (142 m) between gridded values and previous German/Japanese radar data is inevitable result of the smoothing (P4L11). I am afraid that this grid size gives a false impression that the bed topography is known better than we actually know. No details are*

*given about smoothing or data interpolation. This is a key aspect of the final product presented here, however.*

The reviewer raises two important but separate issues: 1) is the kriging routine robust, i.e., is there a risk that topographic features in the thickness maps are artefacts from the interpolation? and 2) do we know the details of the topography in high enough resolution to capture variations over 500m spatial scales?

Issue #1: We consider it unlikely that artefacts have been introduced in the final gridded product by the kriging routine. While radar profiles indeed are separated by 10 km, the along track resolution is very high. Thus, we have detailed geostatistical information on the behaviour of the observations. We assume that this information may be applied to the area as a whole, given that we have no reason to suspect a directional anisotropy in the data. We test this hypothesis further in two ways.

Firstly, we construct 1km, 5km and 10km grids for comparison with the 500m (see below). Even for the coarse resolution maps, the different topographic features are consistent. From the histograms, we also see a very similar distribution of ice thickness values. Indeed, the 500m and 1km maps are almost indistinguishable. We examine the difference between the two maps by linearly interpolating the 1km to 500m resolution. This re-interpolated map should contain fewer details than the original 500m. The mean difference between the two maps is -0.16m with a standard deviation of 8.6m. This confirms that details present in the 500m map are also present in the 1km map.

[Figure]

[Figure]

Secondly, we perform two kriging calculations on a 10km grid, which is on the same spatial scale as the spacing between the radarlines. In one of the calculations, we only include 10% of the data points. The mean difference between the two resulting products is 2.2m with a standard deviation of 18m. Thus, with substantially fewer data points we obtain a very similar result. Performing the same test with two 1km grids returns a mean difference of 2.1m and a standard deviation of 18m.

From these tests, we conclude that our kriging scheme is robust and produces consistent outputs given different resolutions and different amounts of information (data points).

Issue #2: From the investigations above it is evident that we do not gain much additional information by increasing the resolution from 1km to 500m. However, in some regions of our study area, the spacing is indeed 5-10km but in other regions we have substantially more data (including ground-based data). We have clarified this by adding a figure showing the radarlines that were used for constructing the OIR dataset (Fig. 2A). Areas with better data coverage include the area around Dome Fuji where data from several Japanese surveys form a dense grid, and the approach lines to OIR camp and (to some extent) the candidate site northwest of Dome Fuji.

We will take the following actions to address the reviewer's concerns:

We will present our results on a 1km grid but still make the 500m grid available online. In addition, we will present results from a 500m grid in the area around Dome F (shown in Fig. 6) where we have more data and this higher resolution is better justified.

However, we note that changing from 1km to 500m do not change our results or conclusions.

We will provide more information on the kriging routine by adding the following lines to the "Observations and Methods" section:

"The data were subsequently regridded to a  500m and a 1km resolution grid using a kriging interpolation scheme. Based on the experimental variogram, the lag is set to 80km. The experimental variogram is then fitted to an exponential model whose parameters are found by minimising the mean squares difference between the observational variogram and the model. The result of the kriging -- the gridded ice thickness -- is merged with the Bedmap2 topography."

We leave it to the discretion of the editor if the figures showing maps and histograms for different resolution grids should be included in the supplementary material.

**3**

*Very little information is given about lake identification. Nobody can reproduce or assess author's arguments in a meaningful way. Radargrams of at least several lakes should be presented. Do they look like definite lakes, or more like dim lakes? Are all known lakes along the new radar profiles found in this study?*

A figure has been added showing an example of a lake along with layer continuity and the calculated Delta G from the model. In addition, four figures have been added to the appendix also showing different lakes identified in the data as well as layer continuity and ΔG.

Further, lakes identified by Siegert and Fujita have now been added to the figures.

The following lines have been added to the "Basal conditions and internal layering" section

"An example of a radargram with two clear lakes is shown in Fig.5 along with the corresponding layer continuity (cf. Fig. 4B) and the value of *ΔG* from the model (cf. Fig. 3). Four additional radargrams showing examples of lakes and the layer continuity can be seen in the supplementary material.  Previous studies have also identified lakes in the region (Siegert et al., 2005; Fujita et al., 2012) but our analysis adds several new locations. A few of the lakes from the above-mentioned do not appear in our dataset. This might be due to the uncertain geolocation from the old radar data or to the angle of interception of the new radarlines."

**4**

*Sections 2.2 and 2.3 should be better organized. Section 2.2 are written in a way to examine uncertainty of the new data but also to examine old data, which is confusing for me. Section 2.3 presents spatial patterns of the difference between OIR and BEDMAP2, which can be better presented if Sections 2.2 and 2.3 are more efficiently organized/integrated.*

We agree that the description of the uncertainties was unclear and we thank the reviewer for their suggestions on how to improve it. We have reorganised and merged the sections 2.2 and 2.3, and have added more details about the crossover analysis. The section now reads as follows:

"2.2 Results and Uncertainties

[revised manuscript text omitted]

**5**

*Continuity index presented in Fig. 5 should be shown only along the radar profiles and should not interpolate between profiles.*

Fig. 5 (now 5B) has been changed to show continuity index along the lines instead of a gridded version.

**6**

*It is necessary to show radargrams that represent several typical features in this region (e.g., region with high continuity index, lake, transition from the cold base to thawed base) together with continuity index, lake appearance, Delta G, etc. The manuscript shows only limited aspects of the data properties and it is hard for readers to make a comprehensive assessment.*

Four figures have been added to the Appendix showing different kinds of basal conditions and layer continuity. In the section "Basal conditions and internal layering" we add the following sentences:

"An example of a radargram with two clear lakes is shown in Fig. 6 along with the corresponding layer continuity (cf. Fig. 5B) and the value of $\Delta G$ from the model (cf. Fig. 4). Four additional radargrams showing examples of lakes the layer continuity can be seen in the supplementary material."

**7**

*One of main conclusions is that the most promising sites for the oldest ice is in the box I and II. I agree with the authors about this conclusion in general but data should be better presented to fully support this argument. First, why do the authors recommend two separate regions Box I and Box II, instead of one large box that includes both I and II? Second, what defines the boundaries of these boxes? These two are just examples showing the lack of full argument of the main conclusion.*

Thank you for pointing out this issue. We realise that without including a zoom-in on the model results it is difficult for the reader to assess the background for our choices.

We have enlarged the figure and included a zoom-in on the model results. We have also clarified that the two boxes "I" and "II" do not refer to the entire area within the box but rather are meant to indicate that within the boxes, some areas are promising. The section now reads:

"For this region, we favour two areas outlined with white/black rectangles in Fig.7, that each contain two local maxima where $\Delta G$ is elevated. Although the entire region has a value of $\Delta G>5mW\ m^{-2}$, we pinpoint the two areas in particular for two reasons: 1) The value of $\Delta G$ approaches and exceeds $12mW\ m^{-2}$, thus the uncertainty in geothermal heat flux is less important for the robustness of the prediction. 2) The bed topography appears to consist of high plateaus rather than rugged mountain terrain. The second reason is important for layer continuity and to avoid disturbances to the lowermost part of the ice, that would inevitably impact the stratigraphy. The proximity to the ice core site also implies that extending the age--depth information from the ice core to a new drill site would be relatively straightforward."

**8**

*Please enlarge figures, and consider grouping some figures into a single figure (panel a, b, c, ….). it's hard to compare several data properties together in the current form.*

We agree that some of the figures are a bit small, but unfortunately the width of the figures is fixed for The Cryosphere the only option is making all figures the width of two columns. We have changed some of the figures to fill column width. We hope that in the final layout we will have larger figures than the Discussion format. We disagree with the suggestion of merging some of the figures into one. In our opinion, the figures are already quite busy and contain a lot of information. We leave it to the discretion of the editor if this suggestion should be followed.

We have taken the following actions: We have changed the figure showing the modelling result to full two column width and have repositioned the figures showing the hydropotential and the continuity index so that they are next to each other, hopefully making comparison between them easier.

**9**

*Mention data availability.*

Following has been added to the Conclusion

"The gridded thickness data and the thickness picked along the radarlines will be made available through the pangaea.de website."

**Minor points**

*Figure 1: include surface topography (with contours or background color) in both panels. I recommend enlarge both panels for clarity.*

Surface topography has now been included as contours. The figure is now full width in The Cryosphere lay out.

*Figure 2: it's hard to distinguish fully saturated colors and half-transparent. Please modify the figure.*

We assume that this refer to Fig. 3 rather than Fig. 2. The difference between full saturation and half-transparent has been more clear with a red contour line and different colour scheme.

*Figure 3: The candidate sites should have Delta G > 5 mW/m2. Is it better if the colorbar shows Delta G between 5 and 18 (instead of 0 and 18)?*

The colorbar has been changed.

*Figure 4: Consider larger intervals of the contours. Many features are smaller than the radar-profile separations and thus could be artifacts. It's hard for me to see water paths.*

The intervals have been changed and the figure updated to show water paths and lakes more clearly.

**P3L22:** *"Echo-free zone" is defined as the deep ice on which radar returned power decreases in a short vertical distance much more than expected with geometric spreading and englacial attenuation (Fujita et al., 1999). The echo-free zone is not identical with a zone that shows no return signals, which can be caused by limited radar performance.*

The sentence has been changed to

"…was discarded in the calculation to avoid surface noise and the reduced signal from the deepest part of the ice"

**P3L25:** *Is this true? I think that it is the case, but there is no evidence. Probably better to say something like "because the data were collected in late 1980s without modern GPS positioning system."*

We have changed the sentence to make it clearer how we arrived at this conclusion:

"We have not included the Soviet data from the Bedmap2 data in our compilation because the data were collected without Global Positioning System information and thus have a high associated uncertainty in location (pers. comm., Fujita 2017, see also Lythe et al., 2001). "

**P3L29:** *Pulse widths determine a vertical resolution (capability to distinguish two objects separately) but irrelevant to the precision of the range measurement. The observed elevation change is about 0.01% of a typical ice thickness in the study area (2 km), which is below the known accuracy of the radio-wave propagation speed.*

The sentence has been changed to:

"…a reasonable assumption given that elevation changes at Dome Fuji are less than 0.25m/a (e.g., Helm et al., 2014), thus the observed elevation changes are below the known accuracy of the radio-wave propagation speed."

**P4L5**: *show the number of cross-over points analyzed here.*

The following sentence is now included in the new section about results and uncertainties:

"For the GEA-OIR radarlines, 2453 points were found to be crossover points"

**P4L3**: *please mention clearly at the beginning of this paragraph "the uncertainties are analyzed by cross-over analysis of the new datasets and by the comparison of the gridded map product with previous data that were not used in the map product." Many uncertainty analyses are presented here and it's hard to see the overall analytical setup.*

The section about uncertainties have been rewritten and should now be clearer.

**P4L8-9**: *consider showing a histogram of cross-over errors. Also, it makes more sense to show first and third quartile, instead of the standard deviation.*

A histogram of the difference between the GEA-OIR radarlines and the gridded ice thickness map is now included in the manuscript.

**P4L10**: *unclear what negative number means here (which data were subtracted from which).*

This has now been clarified in the new section.

**P4L11**: *radarlines? Confusing because JARE data are ground based.*

Has been changed to radarlines.

*P4L13-19: Comparison with the BEDMAP2 data shows large discrepancy, and the authors argue that it mainly caused by poor positioning of the earlier Soviet data (again: no evidence of the poor positioning). So, these statements are not for determining uncertainties of the newly collected data.*

The uncertainty section has been rewritten to make this clearer.

*P4L24: characteristics of what?*

The section title has been changed to Results and Uncertainties. Thank you for pointing out the lack of clarity.

*P5L15: include the mean and range of geothermal flux in the study area taken from these datasets. I believe that the geothermal flux is a predicted field of this model (rather than an input field) and these geothermal flux datasets are used to derive delta G. So, I think that geothermal flux is mentioned at a wrong place.*

The geothermal heat flux is not predicted by the model. The model predicts the minimum required geothermal heat flux that will cause the bed to reach the pressure melting point and uses the already existing fields to estimate how likely it is that the bed is melting. As such the heat fluxes are inputs.

*P6 Equation: it's confusing to see $G\_min$ and $G\_mean$, as $G\_min$ is a model prediction and $G\_mean$ is the mean of three datasets. How about $G\_min(predicted)$ and $G\_mean(datasets)$ or such?*

The parameters have been changed to $G\_min^{pred}$ and $G\_mean^{data}$

*P6L2: Why is $Sigma\_G < 25 \, mW/m2$ considered as low? Geothernal flux in this region is roughly $50 \, mW/m2$, so I understand that the standard deviation of $25 \, mW/m2$ means that the geothermal flux has +/- 50% uncertainty. In other words, the margin of geothermal flux with which the bed is kept frozen is much less than the geothermal flux dataset uncertainty (i.e. Delta G of $5 \, mW/m2$ is much smaller than the range of the dataset, $25 \, mW/m2$). I understand that the full description of this modeling is given in Van Liefferinge and Pattyn (2013) but still key information should be given here.*

The threshold value of $25 \, mW/m2$ is there to evaluate the dispersion of the GHF data sets. We consider that beyond that value the dispersion is too high between data sets and then we cannot constrain the GHF well enough (based on the mean value of GHF and Sigma of the whole interior ice sheet). The threshold value of 5 for ΔG is there to restrict our choice of oldest ice sites and improve our confidence in these sites. However, it is arbitrary chosen.

We have added to the section:
 "These two threshold values are chosen according to mean geothermal heat flux values for the whole interior Antarctic Ice Sheet. The values restrict the dispersion of geothermal heat flux values between data sets and reduce the likelihood of melting."

*P7L10: don't use several words in interchangeable ways; Lakes, wet areas, water-filled areas, ….*

We have added the following caveat in the section:

"In the following we refer to these areas as ``lakes'' although in our preliminary analysis it is not possible to distinguish between a lake or, for example, water-filled sediments."

And we will refer to wet areas as lakes in the manuscript.

New figure 2

[Figure]

New figure 3

[Figure]

New figure 5

[Figure]

New figure 6

[Figure]

Note: Reviewer's comments are in italics. Changes to manuscript text are in blue.

Reply to reviewers:

**"Glaciological characteristics in the Dome Fuji region and new assessment for 1.5Ma old ice"**

**Reviewer #3**

*This manuscript provides a new glaciological information (ice thickness, bedrock, topography, GHF, basal condition etc.) of the Dome Fuji area on the base on airborne radar surveys conducted during the 2014/15 and 2016/17 Antarctic season. An accurate geophysical survey is prerequisite for any paleoclimatic ice core site selection. The important effort to provide new geophysical survey in a remotest area of East Antarctica must be supported.*

*While the main result is of interest this manuscript suffers of some flaws: it is not clear the use of previous radar survey (Soviet, Japan and bedmap2 compilation) in the new map; the grid cell of 500 m is unsustainable and useless with a survey spacing from 5 to 10 km and relative interpolation of tens km; the information about internal layering depth and age respect to Dome Fuij ice core are not reported anywhere; the description of subglacial lakes detection is very limited and the thickness of melting/no melting threshold is not reported, and it is a crucial point for GHF; as reported by Fisher et al., 2013 it is very important not only find undisturbed ice of 1.5 Myr but also that the ice between 1.2 and 1.5 myr must be enough thick to detect the 40 kyr cycle, a map of the deepest datable isochrones (from DF) is a crucial point to analyse and report.*

We have rewritten several sections of the manuscript, including adding more information on survey spacing, interpolation scheme and lake detection. We have also changed our main result to a 1km map although we retain the 500m resolution in the area around Dome Fuji, where data are more dense.

The inclusion of previous surveys have been made clearer with the addition of a figure (2A) showing all radarlines that are included in the gridded thickness map. However, it is beyond the scope of this study to produce a map of isochrones depth or conduct internal layering analysis beyond the continuity index method.

The observations and methods section new starts with the following paragraph:

[revised manuscript text omitted]

*In detail:*

*Fig 1*

*WGS84 is not a map projection, is a geodetic spatial reference system*

The figure caption has been changed to

The Dome Fuji region in East Antarctica (the left-hand figure is a polar stereographic projection with standard…

*All figure: add elevation contour line, change West with East, Are data plotted only OIR (fig 2 and 5) or GEA-OIR?*

Surface elevation contours have been added to figures 1 and 2, and West has been changed to East in all figures. Fig. 2 has been updated to show all data. The caption of Fig. 5 (now 5B) states "Continuity index of the internal layers in the OIR data."

***Page 3 Line 29-30*** *the elevation change is negligible information, remove.*

We have rewritten part of the paragraph mentioning the elevation change, however, we would like to keep this for the sake of clarity re. uncertainties.

***Page 3 Line 29-33*** *If the Soviet data are not useful due to the position uncertainty why use bedmap2 based on this data? The new grid must be constructed using only the AWI and Japan data, and BEDMAP2 must be used outside the new survey area.*

That is what we have done. With the additional Fig.2A this should be clearer.

***Paragraph 2.2*** *The paragraph must be clarify and analysed only the cross point of radar profile and not the gridded data.*

This section has been rewritten. We use the crossover analysis between radar profiles and the gridded data to examine the uncertainties and smoothing introduced by the kriging scheme. Thus, we believe there is value in doing this analysis.

***Page 4 line 8*** *"parallel" or "perpendicular"?*

Parallel

***Page 4 Line 16-19*** *what is the meaning of "points qualified"? The difference in ice thickness and standard deviation between Soviet and AWI does not appear so large respect to the comparison of GPS position survey of GEA-OIR and AWI-Japan. Please clarify the point and the uncertain in geographic position of Soviet profile.*

The section on uncertainties has been rewritten and it should be clearer that, for example, the standard deviation of the GEA-OIR data compared to the gridded data is 117m while it is 193m for the Soviet data.

The specific sentence re. "points qualified" has been changed to:

"The decision to exclude the Soviet data was partly based on results from a crossover analysis. Comparison between the Soviet radarlines and the OIR and GEA surveys shows that only slightly more than 100 points is within 50m of each other. For these points, the mean difference in ice thickness between the points is -5 m with a standard deviation of 193 m. This larger standard deviation is likely due the poorly resolved bed rock and the large navigational uncertainty in the Soviet radarlines. Considerable differences were also found between Soviet data and ground-based JARE surveys (Fujita, pers. comm., 2016)"

***Figure 2*** *Add the radar profile, why use only OIR profile, instead OIR e GEA? East instead of West*

Figure 2 has been updated to show better which datasets are included in final data product. A separate figure has been added showing a radargram with lakes and continuity index.

***Figure 3*** *Elevation contour, yellow line New Thickness is very far from radar profile (25 km?), is it the gridded area?*

The caption has been changed to:

"…Blue dots show lake locations identified from the radar data, the extent of the gridded OIR thickness map is outlined with a yellow line,.."

***Page 9 Line 2*** *Dome Fuji area is about 1000 km from the coast, and it is not "the most easily accessible region", but the closer to the Dome Fuji Station.*

The sentence has been changed to

"The most promising is also the most easily accessible region from the existing Dome Fuji station:…"

*Fig 6 Add elevation contour line and radar line used, it is not clear (cf Fujita et al., 2012) explain.*

The figure has been changed to include model results from the area. The figure now includes surface contours and the caption reads:

"(A) Ice thickness in the area close to Dome Fuji station in 500 m resolution and (B) Updated predictions of possible Oldest Ice locations (colours) on a 500 m resolution grid compared to the prediction of Van

Liefferinge and Pattyn (2013) outlined in black. The colour scale show the values of ΔG, semi-transparent colours show areas with a threshold horizontal ice-flow velocity of <2m/a, fully saturated colours show areas where the threshold is <1m/a. Blue dots show lake locations identified from the radar data. The white/black boxes indicate the two most favourable Oldest Ice spots according to our analysis."

New figure 2

[Figure]

New figure 6

[Figure]

**List of changes to manuscript**

We now base most of our discussions on results from the 1km gridded ice thickness rather than the 500m data product.

Several sections have been rewritten, notably, Section 2 where the presentation of data and uncertainties have been substantially modified, and more information has been added about data acquisition and radar settings. Other sections have had additional information added and have been rewritten to a lesser extent for clarity.

Several of the figures have been updated, especially Fig. 2 that now include a standard deviation map, Fig. 3 with changed colour scale and axis, a figure of a lake has been included as Fig. 4, and the maps of hydropotential and continuity index are now enlarged. Several figures are also now including surface topography.

---

## Author Response (AR2)

Replies to 2nd review of

**Glaciological characteristics in the Dome Fuji region and new assessment for ``Oldest Ice''**

Reply to reviewer K. Matsuoka

*The revised manuscript is substantially improved from the earlier version and responses to reviewers' comments are mostly satisfactory. Below, I made a few more additional suggestions to improve the manuscript.*

*1. Balance velocities are used as a criteria to define Oldest Ice candidate sites. Are balance velocities re-calculated using the new ice thickness map, or is it adapted from the previous work? Also, how is the shape function (vertical variation of flow speed) assumed? I understand that details are given in Van Liefferinge and Pattyn (2013) but this aspect should be more described. I recommend a new supplemental figure showing the balance velocity (or zoning of < 1 m/a, 1-2 m/a, or > 2 m/a).*

The following has been added in Section 3:

"The input parameters were horizontal ice velocities (assumed to be equal to balance velocities in our region of interest, see suppl. Fig. 7), surface mass balance (van de Berg et al., 2006, van den Broeke 2008) and surface temperature (van den Broeke et al., 2006).  The balance velocities were obtained from the balance fluxes, assuming that the mass of ice flowing out of any area exactly equals the sum of the inflow and the mass accumulated over the area (e.g., Budd and Warner, 1996). The vertically-averaged horizontal velocities thus depend on ice thickness and the prescribed mass balance. The ice is assumed to be internally deforming according to Glen's flow law with exponent *n* = 3 (cf. Pattyn, 2010). Note that the balance velocities are based on the old ice thickness data. The geothermal heat fluxes are from three different studies…"

Suppl. Fig. 7 is attached showing the balance velocities with contours 1m/a and 2m/a outlined.

*2. Delta G are not shown for significant segments of the presented flight paths, e.g. Figs. 5, S3, S4, S5. Figure S5 caption says that Delta G is smaller than the threshold (5 mW/m2), but no explanations are given for the other regions. Why are dG not shown here? If the balance velocity is larger than the criteria, I rather recommend to show dG regardless of the balance velocity but use a different legend (e.g. solid curve if balance velocity is less than 2 m/a, but dashed curve if the speed is larger).*

Thank you for pointing this mistake out. For all figures S3-S6 it should state in the caption that ΔG is not shown where it is below the threshold value of 5mW/m2. Changing the balance velocity threshold does not change this substantially. We will change the caption on the supplementary figures to say

"Example of radargram, continuity index and ΔG values for a radar line. If the value for ΔG is below 5mW/m2 it is not shown in the plot."

*3. Fig. S4 give an interesting insight; dG is ~8 mW/m2 whereas the radar echo shows flat, smooth, and strong echo, a typical indication of the lake. Does it mean that dG has an uncertainty of ~8mW/m2 generally in the study area, or is it more site specific? Fig. S4 shows the data from the furthest location from the dome and I assume that flow speed is rather large there and thus model assumptions are not fully valid. I think that some interpretations of this feature, lake at large dG, give more value to this paper.*

In Fig. S4, we do indeed see two lakes where the model indicates that the geothermal heat flux is above the threshold value. In between the lakes, the value of dG drops below the threshold value.This is caused by the smooth ice thickness map. Most likely, the model sees the area as one large lake rather than two smaller ones. The small depression that contains the lefthand lake is smoothed out in the ice thickness data. We also note that Fig S4A has been shifted a bit compared to 4B and 4C, which makes the difference appear bigger than it actually is.

We add the following on p. 10, line 3

"Four additional radargrams showing examples of lakes and the layer continuity can be seen in the supplementary material. The figures also give insight into how the model resolves small-scale features. In Fig. S4, two small lakes separated by a topographic high are visible in the radargram. In the kriged ice thickness data, this topographic high is smoothed out and to the model the lakes therefore appear as one. "

We update fig S4 so all figures have the same x-axis.

*4. At the bottom of P.8, the authors argue that the western region (76S, 30E) is no longer hopeful. However, the areal difference between the previous and new studies seems not so large; the area reduced about 20% by area but still significant part of the region is still predicted with large dG. I agree with the authors that the flow speed is large and bed is rough there but still I feel that it is too premature to exclude this region from future research. In any case, please write more to support this assessment or tone down the assessment.*

We assume that the reviewer is referring to the paragraph at the bottom of p. 10 where the western region is discussed. We have changed the section to:

"In the western part of this area, values of ΔG are also high, although at a first glance, two properties make it less favourable: [...] the ice is relatively thin in the region (typically less than 2.5 km) which may prove problematic for obtaining an adequate age resolution in an ice core. Furthermore, the distance (several hundreds of kilometres) to Dome Fuji makes it challenging to connect the internal layers to the ice-core chronology. Even so, layer tracing has been achieved across equal or longer distances in this part of Antarctica (e.g., Fujita et al., 2011; Steinhage et al., 2013), and a targeted campaign with radar systems optimised for layer clarity could address this issue."

*5. Please clarify further that temporal resolutions of the prospective ice core are not quantitatively evaluated in this analysis. It is said qualitatively (e.g. thin ice brings low time resolution) but no quantitative analysis is made here and it is left for future research.*

We have modified the beginning of Section 5 to the following:

"Based on the OIR ice thickness dataset, we identify two regions with the best potential for containing Oldest Ice. Our criteria include modelled ΔG values, ice-flow velocities, proximity to the Dome Fuji ice-core

and local ice thickness. For the latter criteria, we make a qualitative assessment essentially assuming that shallower ice thicknesses result in lower time-resolution but we make no quantitative analysis here. The most promising region is also the most easily accessible region from the existing Dome F…"

*Minor points:*

*P4L30-33: I think that it is an interesting observation but the description is too brief to make a solid conclusion. If possible, please calculate the standard deviation for different angle ranges between flight paths (e.g. 0-30, 30-60, 60-90). The standard deviation seems quite large here, as both datasets are collected very recently with high position and ice-thickness accuracy. Bed slope may be an alternative explanation of this large discrepancy.*

There are unfortunately very few lines that intersect each other. Of the ones that do cross other lines within 50m, the flightline that goes from Dome Fuji and towards the west has substantially higher crossover differences compared to most other lines. We have added the following

"We ascribe this difference to geometrical effects of flightline orientation since we observe that radarlines intersecting each other at oblique angles have a larger thickness difference than those that are almost parallel. Of the crossover points where the difference exceeds 100m, one third is situated along the radar line that runs from Dome Fuji to the west, intersecting the other radar lines. "

*P10L13 I think that 35 km is a typo. Note that Figure 6 cation (p11, 2nd line from the bottom) says that it is 2.5 km.*

Yes, thank you for pointing this out. Both are in fact typos. With a data spacing of 35m it should say 3.5km.

**New figures**

Fig. S4

[Figure]

Fig. S7

Balance velocities

[revised manuscript text omitted]